# Strategies of property developers in the context of carbon tax

**Qingzhen Yao**[1]*, **Liangshan Shao**[1,2], **Zimin Yin**[2], **Zhen Wang**[1], **Zhen Chen**[1]

**1** School of Business Administration, Liaoning Technical University, Huludao, 125105, China, **2** School of Management Engineering, Liaoning Institute of Science and Engineering, Jinzhou, 121000, China

* yao0213@126.com

**Data Availability Statement:** All relevant data are within the paper and its Supporting Information files.

**Funding:** The authors gratefully acknowledge the financial supports by the project of social science planning foundation of Liaoning Province of China (Grant No.L22AJY007) and Basic scientific

## Abstract

China needs to guide property developers in actively reducing emissions to reach carbon emission reduction targets and respond to global climate change. A carbon tax is an important policy tool. Still, to establish successful rules to steer property developers' reasonable carbon emission reduction behavior, we must first explore property developers' decision-making mechanisms. This study develops an emission reduction and price game model for property developers under the constraint of a carbon tax. It then applies reverse order induction and optimization methods to identify the game equilibrium solution for property developers. Using the game equilibria, we explore the carbon tax mechanism on emission reduction and property developer pricing strategies. We can derive the following conclusions if the carbon tax policy is not implemented: 1.House prices are related to the substitutability of the two types of competitive property developers. 2.The greater the substitutability, the greater the cost of emission reduction paid by consumers. 3.The game equilibrium carbon emission intensity is the average carbon emission intensity of the housing business. In the situation of enacting a carbon tax, we arrive at the following conclusions: 1.The profits of real estate developers who do not have the advantage of emission reduction continue to decline with the increase of carbon tax. 2. For real estate developers who have the advantage of decreasing emissions, profits declined initially and then increased as the carbon tax rate increased, and only when the carbon tax rate reaches $T^{m1*}$ can they fully leverage the cost advantage and obtain ever-increasing profits. 3.Low tax rates should be adopted by the government at the start of the implementation of the carbon tax policy to provide a buffer time for real estate developers who do not have the advantage of emission reduction costs.

## Introduction

China's $CO_2$ emissions have expanded quickly in recent years, accounting for over 30% of global carbon emissions, making it one of the world's top carbon emitters [1]. In 2018, China's total carbon emissions from the building process were 4.93 billion $tCO_2$, accounting for 51.3% of total national carbon emissions, while residential carbon emissions accounted for 63%. From 2009 to 2018, urban residential power consumption's average annual growth rate was

research projects of Liaoning Provincial Department of Education(LJKMR20221944). The funders had no role in study design, data collection and analysis, decision to publish, or preparation of the manuscript.

**Competing interests:** The authors have declared that no competing interests exist.

11.9%, which is still overgrowing [2]. As a result, reducing carbon emissions in the residential sector has a positive impact on environmental issues.

A carbon tax is a charge paid on the production of carbon dioxide and other greenhouse gas emissions, and it is an effective instrument for reducing carbon emissions [3–6]. Currently, a carbon tax is a straightforward policy tool with little economic impact. Therefore it has been supported in many nations and has produced optimal emission reduction benefits, including Finland, Sweden, Denmark, Japan, and others [7]. Many factors influence the carbon tax system's stability and effectiveness. An ineffective carbon tax may impact the market economy and reduce resource allocation efficiency. As a result, while developing carbon tax policies, carbon emission reduction targets, risks, costs, and economic losses must be considered [8]. Varied industries have different energy usage and profit margins. The real estate industry consumes a lot of energy, invests a lot in reducing emissions, and the cost of reducing emissions varies a lot. To successfully boost property developers' emission reduction efficiency, we must begin at the micro level, investigate their behavioral decision-making mechanisms, and investigate the effect of carbon tax legislation on emission reduction and economic impact.

In China, the implementation of low-carbon housing is hampered by a lack of policy, regulatory, and incentive systems [9–11].Compared to the United Kingdom, the United States, Japan, and other nations, China lacks comprehensive policies to support low-carbon housing development and promotion. Although China has defined a low-carbon building appraisal system, enterprises have yet to completely express their passion for lowering low-carbon housing. A low-carbon housing industry driven by property developers has yet to arise. Currently, cost is the most important consideration for real estate developers, and they are less concerned with the economic benefits of low-carbon housing for consumers. Although low-carbon housing has some additional expenses, it can cut energy usage and provide more significant benefits to customers in the long run. Kats assessed the cost and benefit of low-carbon housing for the first time. The incremental cost of low-carbon housing is 2% of that of conventional housing, but the benefit over the next 20 years is ten times that of traditional housing [12]. Dwaikat studied 17 cases and discovered that the cost increment of more than 90% of low-carbon residential projects ranged between -0.4% and 21% [13]. Liu analyzed the cost increment of 93 low-carbon residential buildings in China, with the cost increment ranging between 29 CNY/m2 and 135 CNY/m2 [14]. The incremental cost of low-carbon housing has decreased dramatically as low-carbon technology has advanced [15]. There is an apparent disparity in the cost premium of low-carbon housing, but it depends on the degree of low-carbon and the technology of residential items.

Property developers perceive cost and profit to be crucial factors to consider when there are no carbon limitations. However, under carbon limitations, property developers' behavior will alter. Many scholars have conducted extensive research on business management behavior under carbon limitations in recent years. Some researchers have conducted modeling research on enterprises' emission reduction decisions. For example, Ma et al. examined the impact of carbon taxes on manufacturers' technology choices by comparing manufacturers' profits under various conditions, taking into account the impact of environmental costs on social welfare [16]. The formulation of the government's carbon price policy was considered in light of this. Lin and Jiainvestigated the benefits of firms in the context of carbon taxation using the dynamic CGE model [17]. They felt that a carbon tax might significantly cut carbon emissions.

As the primary source of carbon emissions, the low-carbon conduct of property developers during the manufacturing and operation process is critical to achieving China's carbon emission reduction goals [18]. However, due to the impact of carbon limits from many factors, such as policy and the market environment, property developers' coping options are uncertain [19]. To fully utilize the government's regulatory role and successfully guide property

developers' appropriate carbon emission reduction behavior, a study on the decision-making mechanism of property developers under carbon limits and their environmental and economic implications is required. The real estate industry consumes much energy. Carbon taxation has more advantages in energy-consuming industries than carbon trading [20]. China's carbon tax program is more favorable to economic development than the carbon trading policy under uncertain conditions after 2015 [20]. There is an optimal tax rate to maximize the effect of emission reduction [21,22].

Carbon trading policies and carbon tax policies are widely utilized worldwide to encourage businesses to reduce emissions reasonably. China launched a national carbon trading market on July 16, 2021. However, carbon trading can only guide firms to actively decrease emissions when the competitive and market supply and demand mechanisms work together. As a result, depending solely on carbon trading to cut emissions has some limitations [23]. Carbon trading must meet two requirements: 1. The enterprise is the object of carbon trading; 2. The enterprise must emit carbon emissions throughout its manufacturing activities. However, for residential buildings, carbon emissions from resident use account for 80% of total carbon emissions, and these carbon emissions are not created by real estate developers but by residents. Real estate developers are the primary source of residential building construction. However, construction companies are responsible for the carbon emissions produced during construction, while real estate developers' actions do not generate carbon emissions. As a result, the carbon trading scheme is ineffective in motivating real estate developers to build low-carbon dwellings.

Real estate developers have two options under the carbon tax policy: 1. construct low-carbon housing; 2. construct conventional housing. How should real estate developers make decisions in a competitive environment to maximize benefits? What is the best real estate developer emission reduction intensity? What effect will the carbon tax have on property developer profits? Based on the aforementioned practical background and research needs, as well as the current research status and development trend in relevant fields, this paper analyzes the decision-making mechanism of property developers under the constraint of the carbon tax from a micro perspective. It investigates the effect and interaction of the carbon tax mechanism from the low-carbon policy environment on emission reduction efforts and the economic effects on property developers. The following are the specific research aims of this paper:

1. Construct a decision-making analysis framework for property developers under the carbon tax to serve as the foundation for decision-makers making company decisions under the carbon tax. Then, assist government authorities in understanding the behavioral features of businesses under the constraints of the carbon tax.

2. To investigate the interaction and stacked influence of carbon emission intensity, profit, and selling price of property developers under the constraint of a carbon tax.

## Models without carbon tax

We assume two types of competing property developers $i, \bar{i} \in \{1, 2\}, i \neq \bar{i}$) in the same market whose products are substitutable. We investigate two types of real estate developers in the same region, without taking regional differences into account in the scope of competition.

The products of the two categories of property developers differ somewhat (One type of real estate developer builds low-carbon houses, while the other type of real estate developer

builds regular houses.), and customer consumption preferences are evenly dispersed. Profit maximization is the decision-making goal of both sorts of property developers.

## Assumption

$s_i$ is the quantity of products purchased by home buyers from property developers;

$\alpha$ is the greatest utility value of residential products to home buyers per square meter;

$f$ is the marginal utility decay rate for each product;

Real estate companies all aim to maximize profits for carbon emission reduction and pricing. Consumers pay $p_1$ and $p_2$ for houses purchased from property developers, and the overall utility function of the house is:

$$X(s_1, s_2) = \sum_{i=1}^{2}(\alpha s_i - \frac{f}{2}s_i^2) - js_1 s_2 - \sum_{i=1}^{2}p_i s_i \tag{1}$$

We can obtain Theorem 2.1.1.

**Theorem 2.1.1:**The optimal quantity of products purchased by home buyers from property developers is:

$$s_i^* = \frac{\alpha}{f+j} + \frac{f}{j^2-f^2}p_i - \frac{j}{j^2-f^2}p_{\bar{i}} \tag{2}$$

**Proof:**

$$\frac{\partial X(s_1, s_2)}{\partial s_1} = \alpha - fs_1 - js_2 - p_1$$

$$\frac{\partial^2 X(s_1, s_2)}{\partial s_1^2} = -f < 0$$

$$\frac{\partial X(s_1, s_2)}{\partial s_2} = \alpha - fs_2 - js_1 - p_2$$

$$\frac{\partial^2 X(s_1, s_2)}{\partial s_2^2} = -f < 0$$

$\frac{\partial^2 X(s_1,s_2)}{\partial s_1^2} < 0, \frac{\partial^2 X(s_1,s_2)}{\partial s_2^2} =< 0$, As a result, when $\frac{\partial X(s_1,s_2)}{\partial s_1} = 0$ and $\frac{\partial X(s_1,s_2)}{\partial s_2} = 0$, $X(s_1, s_2)$ get the maximum size.

Solve the following equations:

$$\begin{cases} \alpha - fs_1 - js_2 - p_1 = 0 \\ \alpha - fs_2 - js_1 - p_2 = 0 \end{cases}$$

We get:

$$\begin{cases} s_1^* = \frac{\alpha}{j+f} + \frac{f}{j^2-f^2}p_1 - \frac{j}{j^2-f^2}p_2 \\ s_2^* = \frac{\alpha}{j+f} + \frac{f}{j^2-f^2}p_2 - \frac{j}{j^2-f^2}p_1 \end{cases}$$

Theorem 2.1.1 is proved.

To make the solution easier, multiply both sides of (2) by $\frac{f^2 - j^2}{f}$ at the same time (this does not affect the qualitative result reached in this work), such that the property developer's product demand function can be changed into:

$$s_i^* = A - p_i + kp_i \tag{3}$$

$A$ is the size of the market inhabited by property developers ($A = \frac{(f-j)\alpha}{f}$), $k$ is the degree of substitutability of products produced by property developers ($k = \frac{j}{f}$), and $0 < k < 1$. We can obtain $A = (1-k)\alpha$.

As a result, we assume that the market size occupied by property developers is related to the maximum utility value of consumers for acquired houses and inversely proportional to the two real estate product substitution coefficient. If $k$ is close to 0, the two property developers are independent of each other and monopolized in their respective markets. The the two types of residential items cannot be replaced. If $k$ is near to 1, it suggests that the two property developers are similar, the market competes with similar items, and the two types of residential products can be substituted for one another. The greater the $k$, the less product difference there is and the more fierce the market rivalry.

## Cost function based on carbon intensity

In practice, property developers must invest more money to lower the carbon footprint of residential structures. As the intensity of carbon emissions decreases, the difficulty of reducing emissions for businesses increases, as does the marginal cost of emission reduction. Because of technological differences, various firms in the same industry have varying marginal emission reduction costs. Taking the marginal effect and difference in the emission reduction cost of property developers into consideration, the production cost per square meter of low-carbon property developers is:

$$c_i(m_i) = c_0 + d_i(\bar{m} - m_i)^2 \tag{4}$$

$c_0$ is the cost per square meter of dwelling before emission reduction, $m_i$ is the property developer's carbon emission intensity after emission reduction, $d_i$ is the cost coefficient of corporate emission reduction, and $\bar{m}$ is the industrial strength's starting carbon emission. Because the enterprise's carbon emission intensity after emission reduction will not be more than the enterprise's starting carbon emission intensity, $0 \leq m_i \leq \bar{m}$.

If we set $\Delta m = \bar{m} - m_i$, the reduction in carbon emissions per square meter of habitation is $\Delta m$. According to (4), the cost of emission reduction is a quadratic function of emission reduction intensity, implying that as the intensity of emission reduction increases, so does the marginal cost of emission reduction. Environmental economists frequently employ similar modeling [24]. $d_i$ (carbon emission reduction cost coefficient) varies depending on the type of real estate. The greater $d_i$, the greater the expense of emission reduction for property developers. Fig 1 depicts the cost of reducing emissions for property developers.

$c_0$ is normalized to 0 for ease of calculation, however this has no effect on the subsequent conclusions. As a result, the production cost per unit can be stated as follows:

$$c_i(m_i) = d_i(\bar{m} - m_i)^2 \tag{5}$$

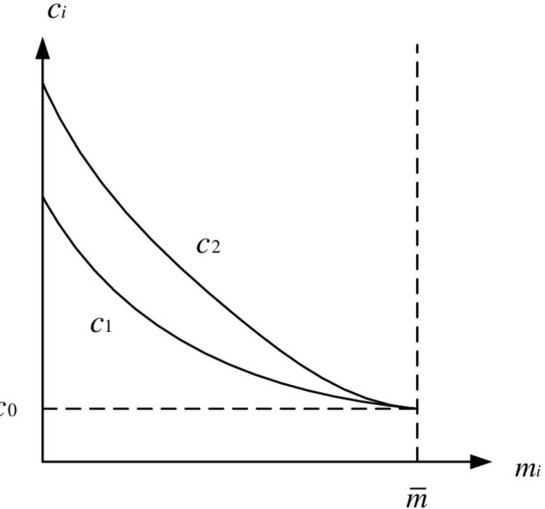

**Fig 1. Change curve of emission reduction cost of property developers.**

## Profit function and game sequence of property developers

**Profit function of property developers.** The unit selling price of the property developer is $p_i$, and the unit cost is $c_i$, so the unit profit of the property developer's unit product is $p_i - c_i$, and the property developer's profit may be represented as:

$$L_i = (p_i - c_i)q_i \tag{6}$$

Substituting (5) into (6) yields:

$$L_i = \left[p_i - d_i(\bar{m} - m_i)^2\right](A - p_i + kp_{\bar{i}}) \tag{7}$$

**Game sequence of property developers.** Assuming that the emission reduction cost coefficient ($d_1, d_2$), emission reduction technology level, and decision-making goals of property developers are all known, the two types of property developers play a game in which they compete for the highest profit.

The game behavior of property producers is examined using Stackelberg game theory. Its core principle is that both parties adopt tactics based on the other's viable strategies to maximize their interests under the other's approach, reaching Nash equilibrium.

The Stackelberg game model can be summed up as $N = \{1, 2\}$ $S_2 = \{q_2 | q_2 = q_2(q_1)\}$, $S_1 = \{q_1 | q_1 \in [0, \infty)\}$, Therefore, the Nash equilibrium can be solved by reverse order induction.

In practice, it takes a long time for property developers to make emission reduction decisions and sell houses. As a result, it makes sense to separate the gaming process into two sections. Based on this, the property developer decision game can be represented as follows:

Step 1: Property developers make emission reduction decisions (such as whether to build low-carbon housing and how to invest in determines the demand for their ($m_1, m_2$);

Step 2: The duopolistic property developers engage in a price game. Property developers price their products based on the assumption that the decision-making information in the first stage is known ($p_1, p_2$).

Ultimately, buyers make decisions based on the low-carbon level and price of the two types of property developers, which also determines the demand for their products $(q_1, q_2)$. The game sequence of property developers is shown in Fig

## Game analysis

We use the retrograde induction method to solve the two types of property developers' Stackelberg game models. The solution steps are reversed from the game order in Fig 2.

Step 1: Assuming that the carbon intensity decisions of the two types of developers are known, solve the price functions of the two types of developers.

Step 2: Substitute the price function into the carbon intensity decision function of the first stage.

Finally, the carbon intensity of the dwelling is incorporated into the price function, and the profits of the two types of property developers are examined.

**Price($P_i^*$) decision of residential products. Theorem 2.4.1**: Given that the carbon intensity $(m_1, m_2)$ of the two property developers in the first stage is known, the property developers' game equilibrium pricing function is:

$$p_i^*(m_i, m_{\bar{i}}) = \frac{A}{2-k} + \frac{2d_i(\bar{m} - m_i)^2}{4 - k^2} + \frac{kd_{\bar{i}}(\bar{m} - m_{\bar{i}})^2}{4 - k^2} \tag{8}$$

**Proof:**

We assume that property developers make carbon emission reduction decisions (such as whether to build low-carbon houses or to invest in technology) and determine the carbon emission intensity of products $(m_1, m_2)$. Then the first-order and second-order partial derivatives of (7) with regard to $p_i$ can be expressed as:

$$\frac{\partial L_i}{\partial p_i} = A + d_i(\bar{m} - m_i)^2 + kp_{\bar{i}} - 2p_i$$

$$\frac{\partial^2 L_i}{\partial^2 p_i} = -2$$

Because of $\frac{\partial^2 L_i}{\partial^2 p_i} < 0$, the profit of the property developer is maximized at $\frac{\partial L_i}{\partial p_i} = 0$. When making judgments, two property developers take each other's decisions into account. As a result, solve the following equations:

$$\begin{cases} A + d_1(\bar{m} - m_1)^2 + kp_2 - 2p_1 = 0 \\ A + d_2(\bar{m} - m_2)^2 + kp_1 - 2p_2 = 0 \end{cases}$$

$$\begin{cases} p_1^* = \frac{A}{2-k} + \frac{2d_1(\bar{m} - m_1)^2}{4 - k^2} + \frac{kd_2(\bar{m} - m_2)^2}{4 - k^2} \\ p_2^* = \frac{A}{2-k} + \frac{2d_2(\bar{m} - m_2)^2}{4 - k^2} + \frac{kd_1(\bar{m} - m_1)^2}{4 - k^2} \end{cases}$$

Theorem 2.4.1 is proved.

The increased cost for enterprises due to emission reduction investment is $d_i(\bar{m} - m_i)^2$, depending on the enterprise's technological parameters and emission reduction intensity. The optimal selling price is to transfer the increased cost to the buyer according to the proportion of $\frac{2}{4-k^2}$, and the price will be raised according to the proportion of $\frac{k}{4-k^2}$ to the competitor's costs. $k \to 0, \frac{2}{4-k^2} \to \frac{1}{2}, \frac{k}{4-k^2} \to 0; k \to 1, \frac{2}{4-k^2} \to \frac{2}{3}, \frac{k}{4-k^2} \to \frac{1}{3}$. As a result, the ideal selling price of a

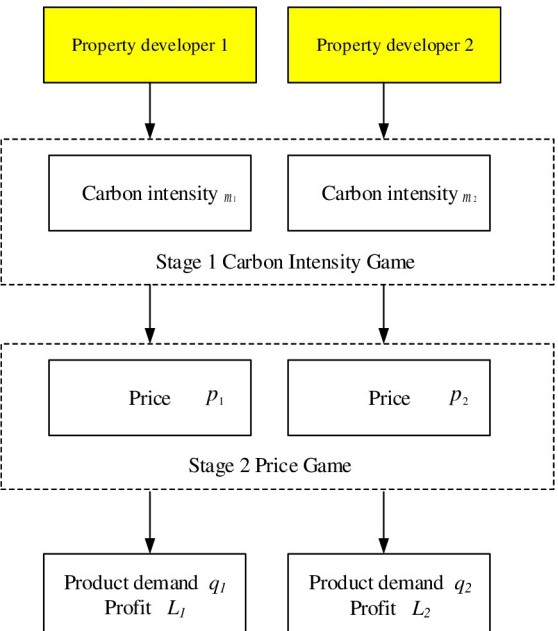

**Fig 2. Game sequence diagram of property developers.**

house is connected to substitutability. The greater the market competition, the higher the cost of emission reductions paid by consumers.

**Decision-making of carbon emission intensity ($m_i^*$). Theorem 2.4.2:** The game equilibrium carbon emission intensity of property developers under the two-stage game model of property developers presented in this paper is:

$$m_i^* = \bar{m} \tag{9}$$

**Proof:**

Substitute Eq (8) into (7) to obtain the property developer's profit function:

$$L_i = \frac{A^2}{(2-k)^2} + \frac{\left[(k^2-2)d_i(\bar{m}-m_1)^2 + kd_i(\bar{m}-m_{\bar{i}})^2\right]^2}{(4-k^2)^2}$$
$$+ \frac{2A(k^2-2)d_i(\bar{m}-m_1)^2 + 2Akd_i(\bar{m}-m_{\bar{i}})^2}{(2-k)(4-k^2)}$$

The profit function($L_i$) of a property developer is derived from the carbon emission intensity($m_i$):

$$\frac{\partial L_i}{\partial e_i} = \frac{4h_i(2-k^2)(\bar{m}-m_i)\left[p_i^* - h_i(\bar{m}-m_i)^2\right]}{4-k^2}$$

In accordance with economic laws, the optimal selling price ($p_i^*$) of the product must be greater than the cost increase ($d_i(\bar{m}-m_i)^2$) caused by emission reduction. We can obtain: if $\bar{m} > m_i$, then $\frac{\partial L_i}{\partial m_i} > 0$; if $\bar{m} < m_i$, then $\frac{\partial L_i}{\partial m_i} < 0$. So, when $\frac{\partial L_i}{\partial m_i} = 0$, property developers can obtain maximum profits.

Let $\frac{\partial L_i}{\partial m_i} = 0$, then $m_i^* = \bar{m}$.

Theorem 2.4.2 is proved.

**Theorem 2.4.3:** The optimal selling price of houses under the model developed in this research is:

$$p_i^* = \frac{(1-k)\alpha}{2-k} \tag{10}$$

**Proof:**

Substitute the game equilibrium carbon emission intensity (9) of property developers into the game equilibrium pricing function (8), and we can get:$p_i^* = \frac{A}{2-k}$.Since $A = \frac{(f-j)\alpha}{f}$, $k = \frac{j}{f}$, so $p_i^* = \frac{(1-k)\alpha}{2-k}$.

Theorem 2.4.3 is proved.

Therefore, in the absence of a carbon tax policy, the optimal product price is related to the degree of substitutability of the two types of products (the degree of market competition).

**Game equilibrium profit of property developers. Theorem 2.4.4:** The game equilibrium profit of property developers is:

$$L_i^* = \frac{(1-k)\alpha}{2-k} \tag{11}$$

**Proof:**

When the game equilibrium carbon emission intensity (9) of property developers is substituted into the game equilibrium price function (8), and we obtain:

$$L_i^* = \frac{A}{2-k}$$

Since $A = (1-k)\,\alpha$, then:

$$L_i^* = \frac{(1-k)\alpha}{2-k}$$

Theorem 2.4.4 is proved.

## Decision function and model construction under carbon tax

The carbon tax is added into the decision-making model of the property developer in this part, and the game equilibrium solution is obtained. Based on this, it examines the decision-making mechanisms of property developers' carbon emission reduction efforts, profits, and product prices in the context of a carbon tax.

Houses emit carbon dioxide during construction and use. We assume, as the principal body and leader of implementation, that property developers will pay a carbon tax for the carbon emissions they produce. The carbon tax rate is $T$, and the carbon tax payable per square meter of dwelling is $Tm_i$. As a result, the profit function of real estate firms subject to carbon tax and subsidy trading legislation is:

$$L_i = \left[p_i - d_i(\bar{m} - m_i)^2 - Tm_i\right](A - p_i + kp_i) \tag{12}$$

Obviously, the carbon tax adds extra cost ($Tm_i$) to the property developer, and the tax rate ($T$) affects the property developer's profit. Under the limits of the carbon tax, the game sequence of property developers is the same as in 2.3.2. We solve the Stackelberg game model using the retrograde induction approach, and the solution stages are consistent with 2.4.

## Carbon intensity, price and profit

*Price decision*

**Theorem 3.1.1:** Given a carbon tax, the price function of property developers is:

$$p_i{}^*(m_i, m_{\bar{i}}) = \frac{A}{2-k} + \frac{2d_i(\overline{m} - m_i)^2 + 2Tm_i}{4-k^2} + \frac{kd_{\bar{i}}(\overline{m} - m_{\bar{i}})^2 + kTm_{\bar{i}}}{4-k^2} \tag{13}$$

**Proof:**

Taking the carbon emission intensity ($m_1$, $m_2$) of property developers as known, the first derivative and second derivative of $p_i$ with respect to the profit ($L_i$)are derived as follows:

$$\frac{\partial L_i}{\partial p_i} = A + d_i(\bar{m} - m_i)^2 + kp_i - 2p_i + Tm_i$$

$$\frac{\partial^2 L_i}{\partial^2 p_i} = -2$$

Since $\frac{\partial^2 L_i}{\partial^2 p_i} < 0$, the property developer's profit is the largest when $\frac{\partial L_i}{\partial p_i} = 0$.

When making decisions, two property developers take each other's decisions into account. As a result, the following equations are developed:

$$\begin{cases} A + d_1(\bar{m} - m_1)^2 + kp_2 - 2p_1 + Tm_i = 0 \\ A + d_2(\bar{m} - m_2)^2 + kp_1 - 2p_2 + Tm_i = 0 \end{cases}$$

We can obtain:

$$\begin{cases} p_1{}^* = \frac{A}{2-k} + \frac{2d_1(\bar{m} - m_1)^2 + 2Tm_i}{4-k^2} + \frac{kd_2(\bar{m} - m_2)^2 + kTm_2}{4-k^2} \\ p_2{}^* = \frac{A}{2-k} + \frac{2d_2(\bar{m} - m_2)^2 + 2Tm_i}{4-k^2} + \frac{kd_1(\bar{m} - m_1)^2 + kTm_1}{4-k^2} \end{cases}$$

Theorem 3.1.1 is proved.

In comparison to the no-carbon-tax condition, the profit function of property developers has increased by two items: $\frac{2Tm_i}{4-k^2}$ and $\frac{kTm_i}{4-k^2}$. This shows that under the given carbon intensity, the carbon tax mechanism will affect the pricing and profits of property developers. On the one hand, the carbon tax paid by property developers is transferred to consumers with a coefficient of $\frac{2}{4-k^2}$. On the other hand, the carbon emission cost of competitors will also be taken into account, and the price of the product will rise by the coefficient of $\frac{k}{4-k^2}$. When $k \to 0$ (decreased product differentiation and increased competition intensity), then $\frac{2}{4-k^2} \to \frac{1}{2}$ and $\frac{k}{4-k^2} \to 0$. When $k \to 1$(Product differentiation increases and competitive intensity decreases), then $\frac{2}{4-k^2} \to \frac{2}{3}, \frac{k}{4-k^2} \to \frac{1}{3}$.

**Decision-making of carbon emission intensity. Theorem 3.1.2:** The equilibrium carbon emission intensity of the two property developers game under the constraint of a carbon tax is:

$$m_i^* = \bar{m} - \frac{T}{2d_i} \tag{14}$$

As a result, the derivative of $L_i$ with respect to $m_i$ is:

$$\frac{\partial L_i}{\partial m_i} = \frac{2(2-k^2)[2d_i(\bar{m}-m_i)-T]}{4-k^2}\left\{\frac{A}{2-k} - \frac{[d_i(\bar{m}-m_1)^2+Tm_i](2-k^2)}{4-k^2} + \frac{k[d_i(\bar{m}-m_i)^2+Te_i]}{4-k^2}\right\}$$

$$= \frac{2(2-k^2)[2d_i(\bar{m}-d_i)-T]}{4-k^2}\left\{p_i^* - [d_i(\bar{m}-m_i)+Tm_i]\right\}$$

According to a common economic phenomenon, a product's selling price is greater than its cost, so $p_i^* - [d_i(\bar{m}-m_i)+Tm_i] > 0$. When $m_i < \bar{m} - \frac{T}{2d_i}, \frac{\partial L_i}{\partial m_i} > 0$; When $m_i > \bar{m} - \frac{T}{2d_i}$时$\frac{\partial L_i}{\partial m_i} < 0$. As a result, When $m_i = \bar{m} - \frac{T}{2d_i}, \frac{\partial L_i}{\partial m_i} = 0$.

$$m_i^* = \bar{m} - \frac{T}{2d_i}$$

Theorem 3.1.2 is proved.

In the case of competition between two oligopolistic property developers, the optimal carbon emission intensity of property developers is jointly determined by the company's emission reduction cost coefficient ($d_i$), the industry's average carbon emission intensity ($\bar{m}$) and carbon tax ($T$), while the carbon emission intensity of competitors emission intensity ($m_{\bar{i}}$) is irrelevant.

**Game equilibrium price. Theorem 3.1.3:** The property developer's game equilibrium price is:

$$p_i^* = \frac{A+Tm_i}{2-k} - \frac{1}{8-2k^2}\left(\frac{T^2}{d_i} + \frac{kT^2}{d_{\bar{i}}}\right) \tag{15}$$

**Proof:**

When we substitute the equilibrium solution (14) of the two property developers' carbon emission intensity game into the equilibrium solution of the price game (13), we get:

$$p_i^* = \frac{A+Tm_i}{2-k} - \frac{1}{8-2k^2}\left(\frac{T^2}{d_i} + \frac{kT^2}{d_{\bar{i}}}\right)$$

Theorem 3.1.3 is proved.

**Equilibrium profit of property developers. Theorem 3.1.4:** Under the constraint of a carbon tax, the game equilibrium profit of property developers is:

$$L_i^* = \frac{\left[A-(1-k)\bar{m}T+\frac{1}{8+4k}\left(\frac{2-k^2}{d_i}-\frac{k}{d_{\bar{i}}}\right)T^2\right]^2}{(2-k)^2} \tag{16}$$

**Proof:**

Substitute the game equilibrium price of property developers (15) and the equilibrium carbon emission intensity of property developers (14) into the profit function of property developers (12), and we can get:

$$L_i^* = \frac{\left[A-(1-k)\bar{m}T+\frac{1}{8+4k}\left(\frac{2-k^2}{d_i}-\frac{k}{d_{\bar{i}}}\right)T^2\right]^2}{(2-k)^2}$$

Theorem 3.1.4 is proved.

## Impact of carbon tax rate on game equilibrium

**The impact of carbon tax rates on carbon emission intensity ($m_i$).   Inference 1:** Under carbon tax constraints, property developers' carbon emission reduction($\Delta m_i$) efforts are proportional to the carbon tax rate($T$), and inversely proportional to the company's emission reduction cost coefficient($d_i$). Property developers' marginal emission reduction cost ($\frac{\partial c_i(\Delta m_i)}{\partial \Delta m_i}$) is proportional to their emission reduction efforts.

**Proof:**

In the game equilibrium state, the emission reduction intensity per unit product of property developers can be expressed as $\Delta m_i^* = \bar{m} - m_i^*$, $m_i^* = \bar{m} - \frac{T}{2h_i}$, then:

$$\Delta m_i^* = \frac{T}{2d_i}$$

Since $c_i(m_i) = d_i(\bar{m} - m_i)^2$, then $c_i(m_i) = d_i \Delta m_i^2$. Correspondingly, the marginal emission reduction cost of property developers is:

$$\frac{\partial c_i(\Delta m_i)}{\partial \Delta m_i} = 2d_i \Delta m_i$$

As a result, property developers' carbon emission reduction efforts are directly proportional to the carbon tax price and inversely proportional to the company's emission reduction cost coefficient.

Inference 1 is established.

As can be shown, the carbon tax can definitely encourage real estate developers to build low-carbon homes, and the lower the cost of emission reduction, the more motivation real estate developers have to actively decrease emissions. As the intensity of emission reduction increases, so does the marginal cost that property developers must bear.

**Impact of carbon tax rate on game equilibrium price.   Inference 2:** Assume that property developer 1 has a cost advantage ($0 < d_1 < d_2$) over property developer 2 in emission reduction. Under the constraint of a carbon tax, the game equilibrium price ($p_1^*, p_2^*$) rises as the carbon tax rate rises, and the rate of rise is reduced, with $p_1^*$ always growing at a lower rate than $p_2^*$.

**Proof:**

(15) shows that $p_1^*$ is a quadratic function about $T$, and the function image is a parabola with a downward opening.

The first and second derivatives of $p_1^*$ with respect to $T$ are as follows:

$$\frac{\partial p_i^*}{\partial T} = \frac{1}{2-k}\left[\bar{m} - \frac{T}{2+k}\left(\frac{1}{d_i} + \frac{k}{2d_{\bar{i}}}\right)\right]$$

$$\frac{\partial^2 p_i^*}{\partial T^2} = -\frac{1}{4-k^2}\left(\frac{1}{d_i} + \frac{k}{2d_{\bar{i}}}\right)$$

Let $\frac{\partial p_i^*}{\partial T} = 0$, the vertex of parabola:

$$T^{mi} = \frac{2d_i d_{\bar{i}}(2+k)\bar{m}}{2d_i + kd_{\bar{i}}} > 0$$

Since $0 < m_i^* < \bar{m}, m_i^* = \bar{m} - \frac{T}{2d_i}$, then $T < 2d_i\bar{m}$. As a result, we can obtain the feasible region of carbon tax as $[0, 2d_1\bar{m}]$.

$$T^{m1} - 2d_1\bar{m} = \frac{2d_1d_2k(1 - \frac{d_1}{d_2})\bar{m}}{2d_2 + kd_1} > 0$$

$$T^{m2} - 2d_1\bar{m} = \frac{4d_1d_2(1 - \frac{d_1}{d_2})\bar{m}}{2d_2 + kd_1} > 0$$

The parabola's vertices are on the right side of the feasible region. Thus, $P_i^*$ is a monotonically increasing function of $T$ in the feasible region $[0, 2d_1\bar{m}]$.

$$\frac{\partial p_1^*}{\partial T} - \frac{\partial p_2^*}{\partial T} = \frac{-(2 - k)(d_2 - d_1)}{(8 - 4k^2)} < 0$$

As a result, $P_1^*$ always grows at a slower rate than $P_2^*$.

Inference 2 is proved.

The game equilibrium price growth rate of real estate developers with cost reduction advantages is lower than that of real estate developers with cost disadvantages when the tax rate rises. This is due to the fact that real estate developers that face cost disadvantages have lowered their emission reduction costs through continual imitation. To further reinforce the cost advantage, however, the advantageous property developers must pay a higher marginal cost.

**Impact of carbon tax on game equilibrium profit. Inference 3:** Assuming that property developer1 has a cost advantage in carbon emission reduction ($0 < d_1 < d_2$), the equilibrium profit of property developer1 ($L_1^*$) decreases at first and then increases as the carbon tax rate increases.

This demonstrates that when the carbon tax rate is low, real estate developers with the cost advantage of emission reduction cannot achieve optimal profits, whereas when the carbon tax rate above a specific threshold, earnings keep rising due to the cost advantage. The cost advantage of emission reduction for real estate developers will arise only when the carbon tax rate hits a particular threshold.

**Proof:**

$$L_i^* = \frac{\left[A - (1 - k)\bar{m}T + \frac{1}{8+4k}\left(\frac{2-k^2}{d_i} - \frac{k}{d_{\bar{i}}}\right)T^2\right]^2}{(2 - k)^2}$$

Let $W_i = \dfrac{A - (1 - k)\bar{m}T + \frac{1}{8+4k}\left(\frac{2-k^2}{d_i} - \frac{k}{d_{\bar{i}}}\right)T^2}{2 - k}$

$$L_i^* = W_i^2$$

$$\frac{\partial W_i}{\partial T} = \frac{-(1 - k)\bar{m}}{2 - k} + \frac{T}{2(4 - k^2)}\left(\frac{2 - k^2}{d_i} - \frac{k}{d_{\bar{i}}}\right)$$

$$\frac{\partial^2 W_i}{\partial^2 T} = \frac{1}{2(4 - k^2)}\left(\frac{2 - k^2}{d_i} - \frac{k}{d_{\bar{i}}}\right)$$

Since $0 < k < 1$, $0 < d_1 < d_2$, then $\frac{2-k^2}{d_1} - \frac{k}{d_2} > 0, \frac{\partial^2 W_1}{\partial^2 T} > 0$. The function graph of $L_i^*$ is a parabola with an upward opening. Let $\frac{\partial W_1}{\partial T} = 0$, the vertex coordinates of the parabola can be obtained:

$$T^{m1*} = \frac{2(1-k)(2+k)\bar{m}d_1 d_2}{(2-k)d_2 - kd_1} > 0$$

Since $0 < m_i^* < \bar{m}$, $m_i^* = \bar{m} - \frac{T}{2d_i}$, then $T < 2d_i\bar{m}$. The feasible domain of carbon tax is $[0, 2d_1\bar{m}]$.

$$T^{m1*} - 2d_1\bar{m} = \frac{-2k\bar{m}d_1 d_2 + 2d_1^2 k\bar{m}}{(2-k^2)d_2 - kd_1} < 0$$

As a result, in the feasible region, $W_1$ decreases first and then increases as $T$ increases. $W_1$ is minimal when $T = T^{m1*}$, and the function graph of $W_1$ is shown in Fig 3.

$$W_{1\min} = \frac{1}{2-k}\left[A - \frac{d_1 d_2(2+k)(1-k)^2\bar{m}^2}{(2-k^2)d_2 - d_1 k}\right] > 0$$

Because $W_{1\min} > 0$, and $W_{1\min}$ is always positive in the feasible region. $W_1 2$ and $W_1$ appear to have similar properties in that $L_1^*$ decreases first and then increases as $T$ increases in the feasible region. As a result, Fig 4 depicts the game equilibrium profit function of property developers.

Figs 3 and 4 show that when the carbon tax rate is low, due to the reduced carbon tax cost, real estate developers prefer to pay carbon tax rather than build low-carbon houses, however in this situation, real estate developers' earnings will continue to drop. It will be unprofitable to continue paying carbon tax as the rate rises. Low-carbon housing technology is being used by real estate developers. Due of its advantages in lowering emissions, market share and earnings will continue to rise.

When the carbon tax rate is less than $T^{m1*}$, it cannot effectively stimulate real estate developers' enthusiasm for building low-carbon housing. When the carbon tax rate exceeds $T^{m1*}$, property developers must use low-carbon technology to construct low-carbon housing. At the same time, the profits of real estate developers increased, creating a virtuous circle. It is worth noting that the carbon tax rate cannot be increased indefinitely, i.e., $T < 2d_1\bar{m}$. Otherwise, the excessive carbon tax will suffocate the industry and create a vicious circle.

**Inference 4:** Assuming that property developer1 has an advantage in terms of emission reduction costs ($0 < d_1 < d_2$), property developer2's equilibrium profit will decrease as the carbon tax rate rises.

This indicates that under the carbon tax policy, the market space of real estate developers who do not have the advantage of emission reduction costs would shrink as the tax rate rises, and earnings will shrink as well.

**Proof:**

$$W_2 = \frac{A - (1-k)\bar{m}T + \frac{1}{8+4k}\left(\frac{2-k^2}{d_2} - \frac{k}{d_1}\right)T^2}{2-k}$$

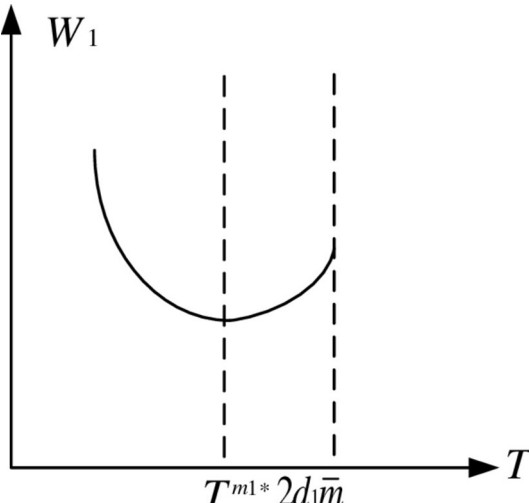

**Fig 3. Function graph of $W_1$.**

The graph of $W_2$ is a parabola with an upward opening. The vertex of the parabola can be found when $\frac{\partial W_2}{\partial T} = 0$:

$$T^{m2*} = \frac{2(1-k)(2+k)\bar{m}d_1 d_2}{(2-k^2)d_2 - kd_1} > 0$$

$$T^{m2*} - 2d_1\bar{m} = \frac{2(2-k^2)(d_1 d_2 - d_1{}^2)\bar{m}}{(2-k^2)d_1 - kd_2} > 0$$

Fig 5 depicts the function image of $W_2$.

Since $W_{2\min} > 0$, $W_{2\min}$ is always positive in the feasible region. We believe that $W_2{}^2$ and $W_2$ have similar properties, that is, $L_2{}^*$ decreases as $T$ increases in the feasible region. As a result, Fig 6 depicts the game equilibrium profit function of property developers.

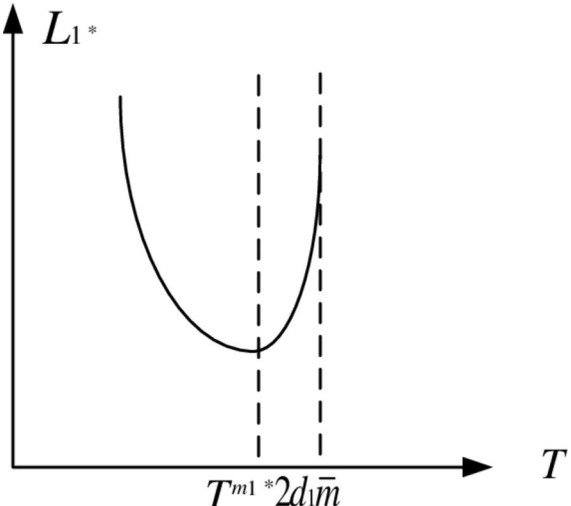

**Fig 4. Function graph of $L_1{}^*$.**

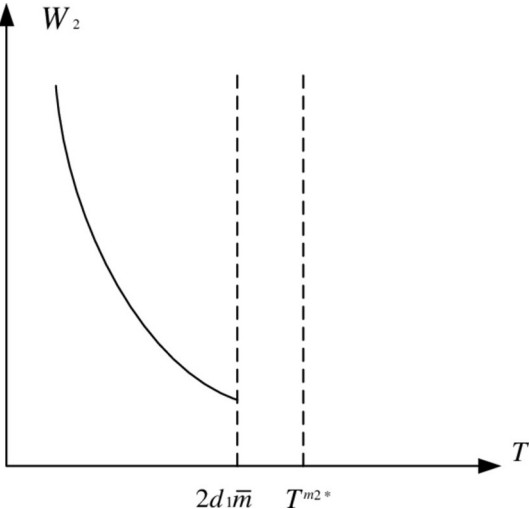

**Fig 5. Function graph of $W_2$.**

Inference 4 is proved.

**Inference 5:** The profit differential between the two property developers is proportional to the square of the carbon tax rate.

**Proof:**

$$L_1{}^* - L_2{}^* = \frac{(1+k)(d_2 - d_1)T^2}{(8+4k)d_1 d_2} > 0$$

Inference 5 is proved.

The profit difference between the two types of real estate developers is proportional to $T^2$, implying that the higher the tax rate, the greater the profit gap between the two. The closer the emission reduction cost coefficient ($d_i$) of the two categories of property developers, the narrower the profit margin. The greater the substitutability ($k$) of these two housing items, the

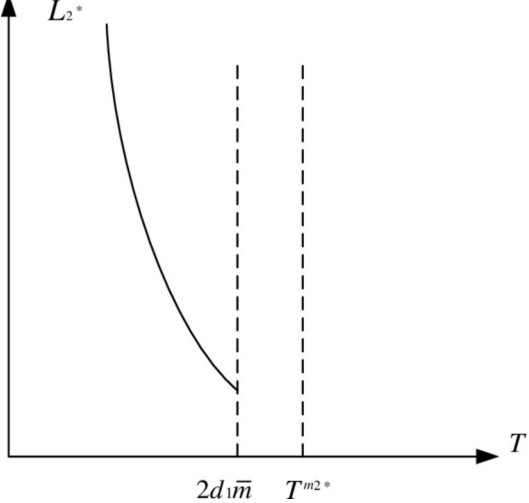

**Fig 6. Function graph of $L_2^*$.**

more competitive real estate developers will be able to produce dwellings with higher cost performance. As a result, the higher the substitutability ($k$), the greater the profit disparity between the two categories of real estate developers.

## Simulation

China's urbanization rate has reached 64.72% as of 2021. Carbon emissions and energy consumption in the construction sector are increasing as China's construction scale expands. The Chinese government's "Opinions on Promoting the Green Development of Urban and Rural Construction" demands that China create high-quality low-carbon buildings and strive for carbon peaking and carbon neutrality in the construction sector. The total energy consumption of residential buildings in China is relatively high. According to China Statistical Yearbook data, urban residential power consumption's average annual growth rate from 2009 to 2018 was 11.9% [2], and it continues to rise rapidly. As a result, China's low-carbon housing development faces enormous challenges. Mao assessed the carbon emissions of 207 residential buildings in Tianjin, and the total carbon emissions from residential structures over their entire life cycle were 2.8tCO$_2$/m2 [25].

Low-carbon housing can be achieved through improving design schemes, using low-carbon building materials, energy-saving equipment, and optimizing production processes. For example, the "Tuanbo Lake Island" project in Tianjin uses passive energy-saving technology such as high-efficiency lighting, high-efficiency air conditioning, curtain wall optimization, and maintenance structure optimization. In addition, the project incorporates low-carbon energy technologies such as solar water heating and photovoltaic power generation based on local conditions to meet users' low-carbon energy needs. Based on a 50-year service period, the economic advantage of energy savings and fossil fuel substitution is 48.2CNY /m$^2$. year, and the low-carbon economic gain is $910CNY / m^2$. As a result, we established $910CNY / m^2$ as the highest utility value per unit of product to customers.

The decrease in emissions caused by low-carbon design solutions, low-carbon equipment procurement, low-carbon construction materials, and low-carbon technological applications varies by different property developers. As a result, determining the emission reduction cost coefficient of property developers is difficult. However, according to climate economics principles, a realistic abatement cost factor is not much different from the carbon tax price. As a result, we set the carbon emission reduction cost coefficient ($d_i$) in this work with reference to the carbon tax rate.

China has not levied a carbon tax, although 29 countries or regions have begun implementing one [26]. Each country has its own carbon tax forms and carbon tax costs. Europe generally has high tax rates, while emerging countries have low tax rates [27]. For example, the tax rate in Sweden is US\$127/tCO$_2$, while the tax rate in underdeveloped countries is between US\$4 and US\$8 [25]. China is still a developing country, therefore, in this paper, $d_1$ is set to 35CNY/t CO$_2$, and $d_2$ is set to 35CNY/t CO$_2$. In fact, we have compared several sets of values assigned to $d_1$ and $d_2$, and found that their variation within a reasonable range will not affect the

**Table 1. Variable details.**

| Variable | unit | property developer 1 | property developer 2 |
|:---:|:---:|:---:|:---:|
| $\overline{m}$ | $tCO_2 / m^2$ | 2.8 | 2.8 |
| $d_i$ | $CNY / tCO_2$ | 35 | 45 |
| $k$ | | 0.5 | 0.5 |
| $\alpha$ | $CNY / m^2$ | 910 | 910 |

conclusions of this paper. $k$ refers to the degree of substitutability of the property developers' products. In this paper's simulation, we assume $k = 0.5$. Table 1 shows the details of the symbols used in this paper.

## Equilibrium price, profit and product substitutability without carbon tax

We simulate the strategy of property developers in the absence of a carbon tax using the parameters given in Table 1. This test confirms the prior analysis results. The substitutability ($k$) of the product influences both the game equilibrium profit ($L_i^*$) and the product selling price ($P_i^*$) of property developers. As the product's substitutability ($k$) increases, the game equilibrium profit ($L_i^*$) and product equilibrium selling price ($P_i^*$) fall. When compared to the product's equilibrium selling price ($P_i^*$), the rate of decline of the game equilibrium profit ($L_i^*$) increases progressively. As shown in Fig 7.

## Impact of carbon tax rate ($T$) on equilibrium carbon intensity ($m_i^*$)

China has not formally imposed a carbon tax, but based on other nations' carbon tax rates, we established a range of carbon tax rates ranging from 0 to 120 (CNY/t $CO_2$). We investigate the impact of different carbon emission reduction cost coefficients($d_i$) on the equilibrium carbon emission intensity of property developers while considering changes in tax rates.

The results reveal that as the carbon tax rate increases, the intensity of carbon emissions ($m_i^*$) falls linearly. Changes in a reasonable interval do not affect the trend of carbon emission intensity ($m_i^*$) decreasing linearly with carbon tax rate increase ($T$). The increase in $d_i$ only slows the fall in $m_i^*$ as $T$ increases. As shown in Fig 8.

## Carbon tax rate ($T$), profit ($L_i^*$) and product substitutability ($k$)

It is assumed that property developer1 has a lower cost of reducing carbon emissions than property developer2. In general, property developer1 earns more profit than property developer2. Figs 9 and 10 demonstrate the trajectory of profit changes in relation to the carbon tax rate. With the increase in the carbon tax, the profit of property developer1 initially declines and then climbs. The profit of property developers2 diminishes as the carbon tax rises. The profit trend of property developers is different when the degree of product substitution and carbon tax are coupled, which is mainly related to the emission reduction cost coefficient ($d_i$) of property developers. As a result, it can only boost firm competitiveness by growing its cost advantage in emission reduction. The simulation results are mutually confirmed with the preceding conclusions.

## Carbon tax rate, profit margin and product substitutability

We investigate the profit gap of the two property developers on the premise of $d_1 = 35CNY / tCO_2$, $d_2 = 45CNY / tCO_2$. The profit margins ($L_1 - L_2$) of the two property developers increase as product substitutability ($k$) increases, while the rate of increase remains essentially constant. $L_1 - L_2$ grows as $k$ increases, and the rate of increase is accelerated.

As a result, we argue that the carbon tax has a greater impact on property developers' profit margins than the degree of product substitutability. Property developers may be motivated to increase their own carbon emission reduction cost advantages in order to profit from the carbon tax policy. As shown in Fig 11.

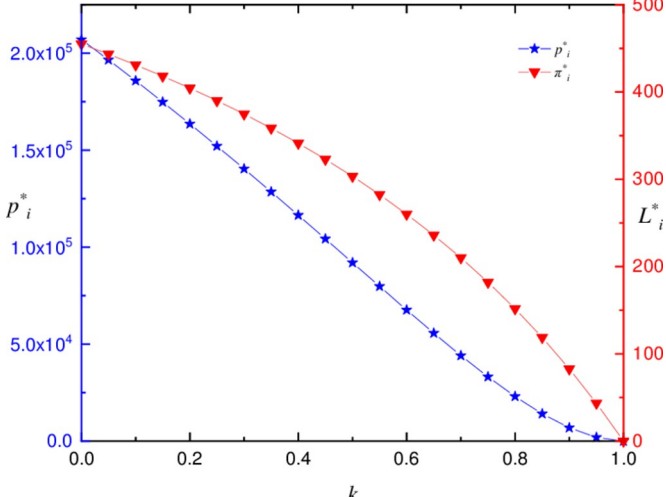

**Fig 7. Profit, product price and product replaceability without carbon tax.**

## Discussion

Residential building carbon emissions have a significant potential to reduce emissions, but the mechanism through which the government can urge property developers to reduce emissions actively is still unclear. We investigate the behavioral decision-making mechanism and its environmental and economic effects of enterprises under the constraint of carbon tax from a micro perspective, which is critical for the government to formulate reasonable carbon emission reduction policies, achieve energy conservation and emission reduction, and mitigate global climate change.

Modeling enterprise operation decision-making under the constraint of the carbon tax has recently been a research hotspot for researchers in production and operation management at home and abroad. There have been many valuable research achievements in related domains. However, there is no comprehensive theory on this topic. There are still many unclear aspects

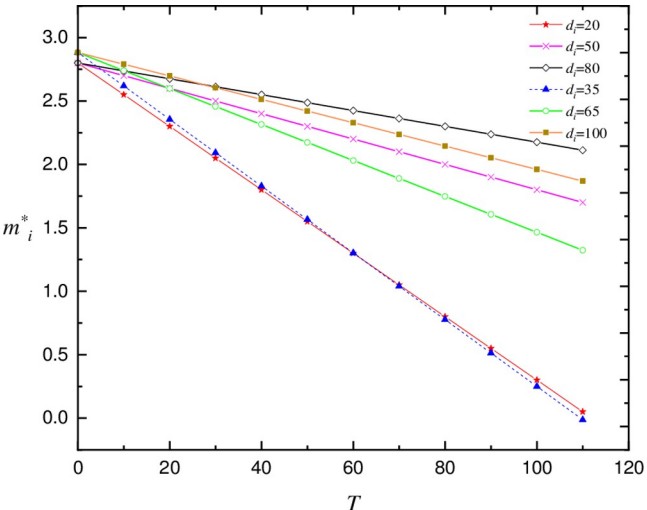

**Fig 8. The effect of $T$ on $m_i^*$.**

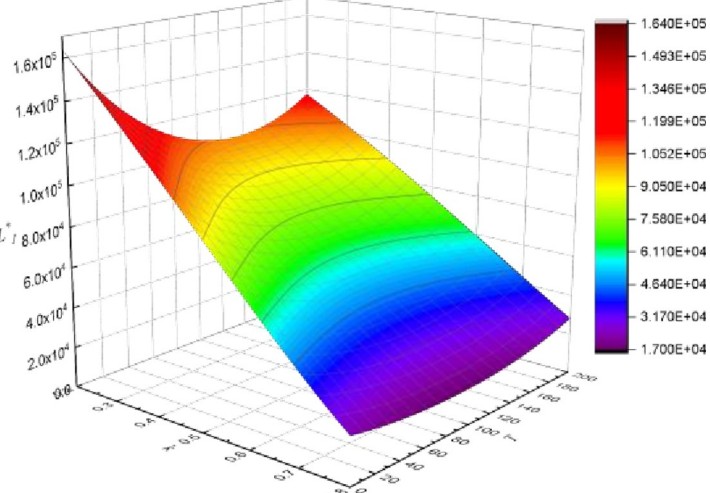

**Fig 9. Carbon tax rate, profit and product substitutability (property developer 1).**

and multi-factor interweaving scenarios in the actual situation that require further in-depth theoretical analysis and modeling study in the analysis framework and research system. As a result, the research effort in this study has both theoretical and practical significance.

Theoretically, this paper builds a strategy and decision analysis framework for property developers under carbon tax constraints, which helps to lay a theoretical foundation for property developers in the field of low-carbon operation management and improve the modeling research system for enterprise operation decision-making under carbon constraints. A two-stage game model of emission reduction and pricing for property developers under carbon tax restrictions is built, which contributes to the enrichment of corporate decision-making models and modeling methodologies under carbon tax constraints.

In terms of application, this paper provides an environmental analysis and strategy selection framework for property developers dealing with carbon tax constraints, which assists government regulators in understanding the behavioral characteristics of property developers

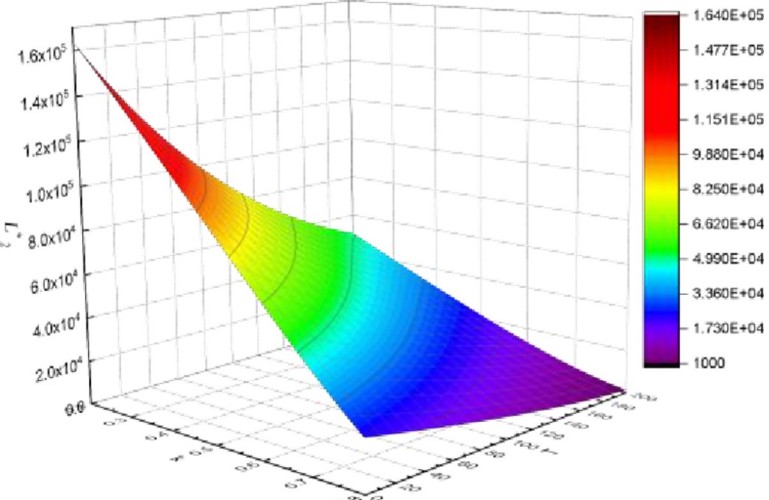

**Fig 10. Carbon tax rate, profit and product substitutability (property developer 2).**

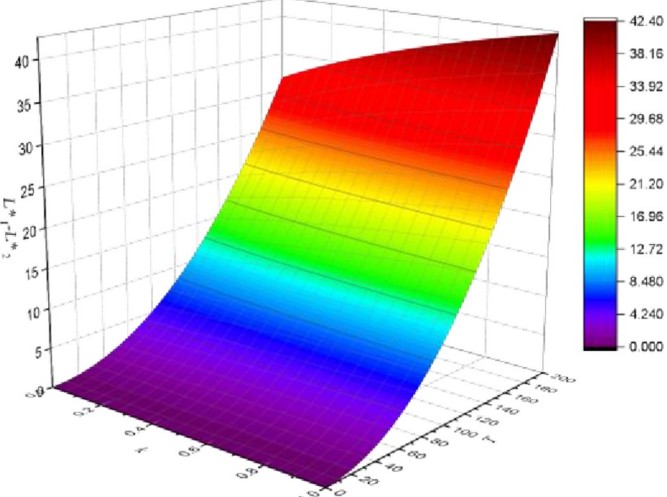

**Fig 11. Carbon tax rate, profit margin and product substitutability.**

dealing with carbon tax constraints. This article examines the company's emission reduction and price decision-making mechanisms under the constraint of a carbon tax, which will assist government regulators in understanding the mechanism of a carbon tax constraint. This paper investigates the joint impact and interaction effect of the carbon tax rate on competitive property developers' carbon emission intensity, product price, and profit. It obtains management inspiration and policy suggestions to help the government guide enterprises' reasonable carbon emission reduction behavior.

Our research builds a two-stage game model of emission reduction and pricing for property developers under the carbon tax trading mechanism based on the actual operation backdrop of the carbon tax mechanism. The game equilibrium solution is utilized to evaluate the emission reduction and pricing decision-making mechanism of property developers under the carbon tax mechanism, and the impact of the carbon tax on emission reduction, product pricing, and property developer equilibrium profits is investigated.

## Conclusion

This research presents a decision-making analysis paradigm for property developers to reduce carbon emissions from a micro perspective. The game models of emission reduction decision-making and pricing of property developers under the conditions of no carbon tax constraint and carbon tax constraint are created based on the symmetry assumption and complete information condition, respectively. Furthermore, the driving effect of carbon tax on property developers is investigated through simulation and analysis, and the interaction mechanism between tax rate, carbon emission reduction intensity, product equilibrium price, and enterprise equilibrium profit is disclosed. Based on this, the following are the key conclusions of this paper:

### No carbon tax constraints

The increased cost for property developers due to emission reduction investment is $d_i(\bar{m} - m_i)^2$, depending on the company's technical conditions and emission reduction intensity. The game equilibrium pricing is to pass on the increased cost to the customer in proportion to $\frac{2}{4-k^2}$, and to raise the price in proportion to $\frac{k}{4-k^2}$ of the competitor's emission reduction

cost. The game equilibrium selling price of residential products is connected to the products' substitutability. The greater the product's substitutability, the greater the market rivalry, and the greater the cost of emission reduction paid by consumers. The game equilibrium profit of property developers is inversely related to the intensity of product rivalry.

## Under the constraints of carbon tax

The carbon tax policy will effect the price and earnings of property developers under the given carbon intensity conditions. On the one hand, the carbon tax ($Tm_i$) paid by property developers is passed on to consumers with an $\frac{2}{4-k^2}$ coefficient. On the other hand, the carbon emission cost ($Tm_i$) of rivals will be considered, and the product's price will rise by the coefficient of $\frac{k}{4-k^2}$.

Property developers' game equilibrium carbon emission intensity is decided by the company's abatement cost coefficient, the industry's average carbon emission intensity, and the carbon tax. It has nothing to do with the competitors' carbon emission intensity. Property developers' carbon emission reduction efforts are directly related to the carbon tax rate and inversely proportional to the company's emission reduction cost coefficient under the carbon tax mechanism.

Based on the the above, we propose the following explanation for the parameter changes in the carbon tax background:

For real estate developers with the advantage of reducing emissions, only when the carbon tax rate exceeds T can they fully exploit the cost advantage and achieve ever-increasing profits.

Under the carbon tax policy, the profits of real estate developers who do not have the advantage of emission reduction continue to decline with the increase of carbon tax. In this case, we recommend that such real estate developers: 1. Increase research into low-carbon technology in order to reduce the cost of emission reduction; 2. Build distinctive houses while attempting to reduce the replaceable degree of houses ($k$).

Low tax rates should be adopted by the government at the start of the implementation of the carbon tax policy to provide a buffer time for real estate developers who do not have the advantage of emission reduction costs. Following that, tax rates should be raised to stimulate the enthusiasm of real estate developers with the benefit of lower emission costs.

## Author Contributions

**Conceptualization:** Qingzhen Yao.

**Data curation:** Qingzhen Yao, Liangshan Shao, Zimin Yin, Zhen Chen.

**Formal analysis:** Liangshan Shao, Zhen Wang.

**Investigation:** Qingzhen Yao.

**Methodology:** Qingzhen Yao.

**Project administration:** Qingzhen Yao.

**Resources:** Qingzhen Yao.

**Supervision:** Qingzhen Yao, Liangshan Shao.

**Validation:** Qingzhen Yao.

**Visualization:** Qingzhen Yao.

**Writing – original draft:** Qingzhen Yao, Zimin Yin, Zhen Chen.

**Writing – review & editing:** Qingzhen Yao, Zhen Wang.

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
