## [Decision Letter · Decision Letter 0]

3 Jan 2023

PONE-D-22-27591Strategies of duopoly property developers in the context of carbon taxPLOS ONE

Dear Dr. Qingzhen Yao,

Thank you for submitting your manuscript to PLOS ONE. After careful consideration, we feel that it has merit but does not fully meet PLOS ONE' s publication criteria as it currently stands. Therefore, we invite you to submit a revised version of the manuscript that addresses the points raised during the review process.

We look forward to receiving your revised manuscript.

Kind regards,

Bo Huang

Academic Editor

PLOS ONE

Journal Requirements:

2. Please amend the manuscript submission data (via Edit Submission) to include author “Zimin Yinb  Zhen Wanga  Zhen Chena”

Reviewers' comments:

Reviewer's Responses to Questions

**Comments to the Author**

1. Is the manuscript technically sound, and do the data support the conclusions?

Reviewer #1: Yes

Reviewer #2: Yes

2. Has the statistical analysis been performed appropriately and rigorously? 

Reviewer #1: Yes

Reviewer #2: N/A

3. Have the authors made all data underlying the findings in their manuscript fully available?

Reviewer #1: Yes

Reviewer #2: Yes

4. Is the manuscript presented in an intelligible fashion and written in standard English?

Reviewer #1: Yes

Reviewer #2: Yes

5. Review Comments to the Author

Reviewer #1: Comment a:

The author draws four conclusions in the case of activating a carbon tax, but the first two conclusions seem to have little intention. Even without corresponding model deduction, it is easy to get relevant opinions.

Comment b:

The foreshadowing in the introduction part is too wordy and does not lead to the research questions of this paper concisely and clearly, so it is not readable. In addition, the "To successfully boost property developers' emission reduction potential" used by the author has problems. The potential should be a relatively objective existence and should be to improve the efficiency of emission reduction.

Comment c:

Where the model is set up, the author asserts two types of competing property developers in the same market where products are substitutable, and refer to them as duopoly competitive property developers This duopoly competition hypothesis deviates greatly from the actual characteristics of China's real estate market. China's real estate market has experienced explosive development, and a large number of real estate enterprises have emerged. Even in the county level cities, the competitors in the real estate market are also very diversified, rather than duopoly competition. In the real estate market of cities above county level, it is a multi game between a large number of high-end brands. If the author focuses on the real estate market below the county level (towns and townships), more houses in relevant markets are built by residents themselves, which has little relevance to the research topic and has no practical significance. At the same time, the Chinese government has actively carried out anti-monopoly activities and has also issued anti-monopoly laws. The author's duopoly competition hypothesis will not appear in the future Chinese real estate market, and the relevant research results have limited significance. In addition, the author studies the real estate duopoly competition game in the context of carbon reduction. However, it is difficult to find a country whose real estate market conforms to the characteristics of duopoly competition among the major carbon emitting countries.

Comment d:

The author advocates the duopoly game model of the real estate market constructed by themselves. However, the relevant model settings do not show the particularity of the real estate market. Just changing the product name to the square number of real estate does not reflect the particularity of the real estate market. We should further consider the impact of high circulation demand, capital chain demand, government price limits, purchase restrictions and other factors of real estate developers.

Comment e:

The real estate market in different regions of China is very different, and the simulation part can be used for regional comparative analysis.

Reviewer #2: Dear authors

This study analyses the impact of environmental taxes on firms' pollution reduction behaviour by utilising the Stackelberg model. After deriving the equilibrium analytically, a simulation analysis with the Chinese economy in mind reveals the impact of environmental taxes on pollution emissions and firm profits. As China has not officially introduced an environmental tax, the results of this study may provide useful insights for policy makers. However, while the equilibrium of the economic model is carefully demonstrated, its econometric intuition is not well explained. Instead, the intuition is explained by focusing on the results of a simulation analysis assuming specific parameter conditions. The explanation of the intuition for the analytical results further enhances the generality of this analysis. The results further increase the persuasiveness of the simulation analysis. The authors should make an effort to increase the generality of their analysis for the acceptance of this study in this journal.

Please see the attached file for more details.

Sincerely.

6. PLOS authors have the option to publish the peer review history of their article (what does this mean?). If published, this will include your full peer review and any attached files.

Reviewer #1: **Yes: **JiaChen Fan

Reviewer #2: No

---

## [Author Response · Author response to Decision Letter 0]

23 Feb 2023

Dear editor and Reviewers:

Thank you for taking the time to process the submission of our original paper entitled Strategies of duopoly property developers in the context of carbon tax. Thank you very much for your recognition of our manuscript. We have carefully revised the manuscript according to your suggestions once more. We hereby resubmit the revised manuscript and hope that all corrections are satisfactory. Please feel free to contacts with any questions and we look forward to your decision.

Sincerely yours,

Authors

 

Reviewer #1 Comment a:

The author draws four conclusions in the case of activating a carbon tax, but the first two conclusions seem to have little intention. Even without corresponding model deduction, it is easy to get relevant opinions.

Response

These two conclusions appear straightforward. We derive the quantitative relationship between variables in the manuscript.

For conclusion 1, “The optimal carbon emission intensity of property developers is defined jointly by the company's emission reduction cost coefficient, the industry's average carbon emission intensity, and the carbon tax, with no regard for competitors' carbon emission intensity.” This conclusion describes the best intensity of emission reduction. Furthermore, we develop the relationship between the ideal emission reduction intensity and the variable, that is, , in the manuscript.

For conclusion 2,“Under the carbon tax mechanism, property developers' carbon emission reduction efforts are directly proportionate to the carbon tax rate and inversely proportional to their emission reduction expenses.” We derive not only the qualitative relationship between emission reduction intensity, cost coefficient, and tax rate in the paper, but also their quantitative relationship, that is, .

Reviewer #1 Comment b:

The foreshadowing in the introduction part is too wordy and does not lead to the research questions of this paper concisely and clearly, so it is not readable. In addition, the "To successfully boost property developers' emission reduction potential" used by the author has problems. The potential should be a relatively objective existence and should be to improve the efficiency of emission reduction.

Response

Thank you for your suggestion. The introduction is really not concise. We revised the introduction and removed the superfluous section. Details can be found in the revised manuscript.

There are indeed problems with the "o successfully boost property developers' emission reduction potential" we use. We have corrected it to " To successfully boost property developers' emission reduction efficiency ".

Reviewer #1 Comment c:

Where the model is set up, the author asserts two types of competing property developers in the same market where products are substitutable, and refer to them as duopoly competitive property developers This duopoly competition hypothesis deviates greatly from the actual characteristics of China's real estate market. China's real estate market has experienced explosive development, and a large number of real estate enterprises have emerged. Even in the county level cities, the competitors in the real estate market are also very diversified, rather than duopoly competition. In the real estate market of cities above county level, it is a multi game between a large number of high-end brands. If the author focuses on the real estate market below the county level (towns and townships), more houses in relevant markets are built by residents themselves, which has little relevance to the research topic and has no practical significance. At the same time, the Chinese government has actively carried out anti-monopoly activities and has also issued anti-monopoly laws. The author's duopoly competition hypothesis will not appear in the future Chinese real estate market, and the relevant research results have limited significance. In addition, the author studies the real estate duopoly competition game in the context of carbon reduction. However, it is difficult to find a country whose real estate market conforms to the characteristics of duopoly competition among the major carbon emitting countries.

Response

This is a very valuable suggestion. As you mentioned, the term "duopoly real estate developers" is improper. We replaced "duopoly real estate developers" to "two types of real estate developers" in the revised manuscript.

We assume that these two categories of real estate developers' real estate developers have similar characteristics, such as brand effect and scale. Yet, one type of real estate developer (real estate developer A) constructs low-carbon dwelling, whereas the other (real estate developer B) constructs regular housing. In this scenario, the two are engaged in a game. The strategy of real estate developer A will be adjusted at any time in accordance with the plan of real estate developer B. Similarly, real estate developer B will modify its plan in response to real estate developer A's strategy.

Ultimately, the two types of real estate developers' tactics will reach a satisfactory balance. The paper analyzes the enterprise emission reduction and pricing decision-making mechanism under the carbon emission trading mechanism, and investigates the impact of the carbon emission trading price on the emission reduction efforts, product pricing, and balanced profits of real estate developers based on the game balance results.

This article's research topic is commercial housing in cities and towns. Most dwellings in China's rural areas are self-built by residents, which is outside the scope of our research.

Reviewer #1 Comment d:

The author advocates the duopoly game model of the real estate market constructed by themselves. However, the relevant model settings do not show the particularity of the real estate market. Just changing the product name to the square number of real estate does not reflect the particularity of the real estate market. We should further consider the impact of high circulation demand, capital chain demand, government price limits, purchase restrictions and other factors of real estate developers.

Response

The particularity of the real estate industry:

Carbon trading policies and carbon tax policies are widely utilized worldwide to encourage businesses to reduce emissions reasonably. China launched a national carbon trading market on July 16, 2021. However, carbon trading can only guide firms to actively decrease emissions when the competitive and market supply and demand mechanisms work together. As a result, depending solely on carbon trading to cut emissions has some limitations (Wu et al., 2014). Carbon trading must meet two requirements: 1. The enterprise is the object of carbon trading; 2. The enterprise must emit carbon emissions throughout its manufacturing activities. However, for residential buildings, carbon emissions from resident use account for 80% of total carbon emissions, and these carbon emissions are not created by real estate developers but by residents. Real estate developers are the primary source of residential building construction. However, construction companies are responsible for the carbon emissions produced during construction, while real estate developers' actions do not generate carbon emissions. As a result, the carbon trading scheme is ineffective in motivating real estate developers to build low-carbon dwellings.

Based on the aforementioned practical background and research needs, as well as the current research status and development trend in relevant fields, this paper analyzes the decision-making mechanism of property developers under the constraint of the carbon tax from a micro perspective. It investigates the effect and interaction of the carbon tax mechanism from the low-carbon policy environment on emission reduction efforts and the economic effects on property developers.

The financial chain, government price controls, and other factors all have an impact on real estate. The popularity and sales of real estate developers impact their profitability. We assume in this analysis that the capital chain, government price limit, and purchase limitation factors are not treated as competitive variables among real estate developers in the same city and real estate developers from other cities. We'll investigate into "whether to construct low-carbon housing" as the sole competitive factor.

Reviewer #1 Comment e:

The real estate market in different regions of China is very different, and the simulation part can be used for regional comparative analysis.

Response

Indeed, real estate prices vary widely across China. The cost of land varies substantially among places, yet the reason for this variation is that the cost of land changes greatly. For example, the land price for housing construction in Beijing is 50000-100000CNY/m2, however in some Chinese counties, the land price for housing development is 500-5000CNY/m2. Nonetheless, the cost of building a house does not differ greatly. Real estate developers' development of low-carbon dwellings will raise construction costs, which have nothing to do with land costs.

We investigate two types of real estate developers in the same region, without taking regional differences into account in the scope of competition.

However, the difference in residents' willingness to purchase low-carbon housing in different regions is a topic worth researching. This work will be investigated further in future research.

 

Reviewer #2 Question 1

The significance of this study is not very clear. A paragraph should be added in the introduction that clearly explains what the key research questions are that previous research analyses have missed and how this study seeks to meet those questions. In doing so, the significance of the study will be clearer if it is clarified how the setting differs from existing theoretical models, and how these differences affect the results.

Response

This is an excellent suggestion. In accordance to your recommendation, we have added the following content in the introduction: 1. The existing carbon trading policy does not apply to the real estate business, however the carbon tax policy does. 2. The issues to be addressed in this study. As follows:

Carbon trading policies and carbon tax policies are widely utilized worldwide to encourage businesses to reduce emissions reasonably. China launched a national carbon trading market on July 16, 2021. However, carbon trading can only guide firms to actively decrease emissions when the competitive and market supply and demand mechanisms work together. As a result, depending solely on carbon trading to cut emissions has some limitations. Carbon trading must meet two requirements: 1. The enterprise is the object of carbon trading; 2. The enterprise must emit carbon emissions throughout its manufacturing activities. However, for residential buildings, carbon emissions from resident use account for 80% of total carbon emissions, and these carbon emissions are not created by real estate developers but by residents. Real estate developers are the primary source of residential building construction. However, construction companies are responsible for the carbon emissions produced during construction, while real estate developers' actions do not generate carbon emissions. As a result, the carbon trading scheme is ineffective in motivating real estate developers to build low-carbon dwellings.

Real estate developers have two options under the carbon tax policy: 1. construct low-carbon housing; 2. construct conventional housing. How should real estate developers make decisions in a competitive environment to maximize benefits? What is the best real estate developer emission reduction intensity? What effect will the carbon tax have on property developer profits?

Reviewer #2 Question 2

Section 3.2, where the comparative statics analysis is performed, is the key section comprising the main results of this study. In doing so, the authors only describe their results in each Inferences, with scant explanation of their economic intuitions. In the absence of this explanation, it is difficult to ascertain what mechanisms are at work behind the simulation analysis in Chapter 4. The authors should follow their proof with a careful explanation of their intuition about Inferences.

Response

Thank you very much for your advice. We have amended the paper to provide the economic explanation for the conclusion in Section 3.2, as suggested by reviewer.

Inference 1：

Under carbon tax constraints, property developers' carbon emission reduction( ) efforts are proportional to the carbon tax rate( ), and inversely proportional to the company's emission reduction cost coefficient( ). Property developers' marginal emission reduction cost ( ) is proportional to their emission reduction efforts.

Explanation of Inference 1：

As can be shown, the carbon tax can definitely encourage real estate developers to build low-carbon homes, and the lower the cost of emission reduction, the more motivation real estate developers have to actively decrease emissions. As the intensity of emission reduction increases, so does the marginal cost that property developers must bear.

Inference 2: 

Assume that property developer 1 has a cost advantage ( ) over property developer 2 in emission reduction. Under the constraint of a carbon tax, the game equilibrium price ( ， ) rises as the carbon tax rate rises, and the rate of rise is reduced, with always growing at a lower rate than .

Explanation of Inference 2：

The game equilibrium price growth rate of real estate developers with cost reduction advantages is lower than that of real estate developers with cost disadvantages when the tax rate rises. This is due to the fact that real estate developers that face cost disadvantages have lowered their emission reduction costs through continual imitation. To further reinforce the cost advantage, however, the advantageous property developers must pay a higher marginal cost.

Inference 3: 

Assuming that property developer1 has a cost advantage in carbon emission reduction ( ), the equilibrium profit of property developer1( ) decreases at first and then increases as the carbon tax rate increases.

Explanation of Inference 3：

This demonstrates that when the carbon tax rate is low, real estate developers with the cost advantage of emission reduction cannot achieve optimal profits, whereas when the carbon tax rate above a specific threshold, earnings keep rising due to the cost advantage. The cost advantage of emission reduction for real estate developers will arise only when the carbon tax rate hits a particular threshold.

Inference 4: 

Assuming that property developer1 has an advantage in terms of emission reduction costs ( ), property developer2's equilibrium profit will decrease as the carbon tax rate rises.

Explanation of Inference 4：

This indicates that under the carbon tax policy, the market space of real estate developers who do not have the advantage of emission reduction costs would shrink as the tax rate rises, and earnings will shrink as well.

Inference 5: 

The profit differential between duopoly the two property developers is proportional to the square of the carbon tax rate.

Explanation of Inference 5：

The game equilibrium profit difference between the two types of real estate developers is proportional to , implying that the higher the tax rate, the greater the profit gap between the two. The closer the emission reduction cost coefficient ( ) of the two categories of property developers , the narrower the profit margin. The greater the substitutability ( ) of these two housing items, the more competitive real estate developers will be able to produce dwellings with higher cost performance. As a result, the higher the substitutability ( ), the greater the profit disparity between the two categories of real estate developers.

Reviewer #2 Question 3

In Figures 3 and 4, there is a non-monotonic relationship between the profit of the property developer and the environmental tax. This relationship arises because the environmental tax creates a profit-raising effect in addition to a profit-decreasing effect. It should be intuitively explained that through these effects a non-monotonic relationship is created.

Response

Figures 3 and 4 show that when the carbon tax rate is low, due to the reduced carbon tax cost, real estate developers prefer to pay carbon tax rather than build low-carbon houses, however in this situation, real estate developers' earnings will continue to drop. It will be unprofitable to continue paying carbon tax as the rate rises. Real estate developers with the advantage of reducing emissions continue to increase their market share and profits by building low-carbon houses.

Reviewer #2 Question 4

The non-monotonic relationship between profit and environmental taxes generates a threshold environmental tax. It should be proved, if possible, by what parameter changes this threshold increases or decreases. Using the results as an Inference, intuitively explain when environmental taxes become more profitable. The results would provide useful policy recommendations.

Response

In Chapter 3.2.3, we used strict mathematical deduction to prove that the abscissa of the vertex of the parabola is , i.e., is the carbon tax threshold.

 is determined by the substitutability ( ) of the two types of products, the industry's initial carbon emission intensity ( ), and the real estate developer's emission reduction cost coefficient ( ). is proportional to , but and have a nonlinear effect on .

When the carbon tax rate is less than , it cannot effectively stimulate real estate developers' enthusiasm for building low-carbon housing. When the carbon tax rate exceeds , property developers must use low-carbon technology to construct low-carbon housing. At the same time, the profits of real estate developers increased, creating a virtuous circle. It is worth noting that the carbon tax rate cannot be increased indefinitely, i.e., . Otherwise, the excessive carbon tax will suffocate the industry and create a vicious circle.

Reviewer #2 Question 5

In Inference 4, the authors should explain why the effect of environmental taxes to increase profits disappears when firms have an advantage with regard to emission reduction costs.

Response

This demonstrates that when the carbon tax rate is low, most real estate companies in the market choose to pay the carbon tax rather than build low-carbon housing. Even if property developers have a cost advantage of reducing emissions, in the face of market competition, they will choose to pay carbon tax rather than build low-carbon housing. As a result, when the carbon tax rate is low, real estate developers' advantages of lower emission costs cannot be used, and profits are reduced. When the carbon tax rate exceeds a certain threshold, the profit will continue to rise due to the cost advantage.

Reviewer #2 Question 6

In Inference 5, the authors should explain what parameter changes would reduce or increase this difference.

Response

The profit differential between the two types of developers is as follows:

The profit difference between the two types of real estate developers in the game is proportional to , which means that the higher the tax rate, the greater the profit difference between the two. The smaller the profit gap, the closer the emission reduction cost coefficient（ ）of the two types of property developers is. The higher the substitutability (k) of the two types of residential products, the better the performance-price ratio of the houses constructed by the advantageous real estate developers. As a result, the greater the substitutability (k), the greater the profit gap between the two types of real estate developers.

Reviewer #2 Question 7

Based on the results of Inference 1-5 and the parameter changes, the authors should clarify under what economic conditions environmental taxes have a profit-increasing effect.

Response

Based on the the above, we propose the following explanation for the parameter changes in the carbon tax background: 

For real estate developers with the advantage of reducing emissions, only when the carbon tax rate exceeds T can they fully exploit the cost advantage and achieve ever-increasing profits.

Under the carbon tax policy, the profits of real estate developers who do not have the advantage of emission reduction continue to decline with the increase of carbon tax. In this case, we recommend that such real estate developers: 1. increase research into low-carbon technology in order to reduce the cost of emission reduction; 2. build distinctive houses while attempting to reduce the replaceable degree of houses ( ).

Low tax rates should be adopted by the government at the start of the implementation of the carbon tax policy to provide a buffer time for real estate developers who do not have the advantage of emission reduction costs. Following that, tax rates should be raised to stimulate the enthusiasm of real estate developers with the benefit of lower emission costs.

Reviewer #2 Question 8

The additional explanations provided by this revision could be summarised as policy recommendations in the conclusion.

Response

This is an excellent suggestion. We revised the conclusion in the rework manuscript based on this suggestion. Details can be found in the revised manuscript.( Conclusion and Abstract section of the manuscript)

Reviewer #2 Question 9

In the proof of Theorem 2.1.1, there is a formula with a small font size of 0 and a formula containing且.

Response

Thank you for your timely correction. We have made changes in the revised manuscript.

---

## [Decision Letter · Decision Letter 1]

12 Mar 2023

Strategies of property developers in the context of carbon tax

PONE-D-22-27591R1

Dear Dr. Qingzhen Yao,

We’re pleased to inform you that your manuscript has been judged scientifically suitable for publication and will be formally accepted for publication once it meets all outstanding technical requirements.

Kind regards,

Bo Huang

Academic Editor

PLOS ONE

Additional Editor Comments (optional):

Reviewers' comments:

Reviewer's Responses to Questions

**Comments to the Author**

1. If the authors have adequately addressed your comments raised in a previous round of review and you feel that this manuscript is now acceptable for publication, you may indicate that here to bypass the “Comments to the Author” section, enter your conflict of interest statement in the “Confidential to Editor” section, and submit your "Accept" recommendation.

Reviewer #1: All comments have been addressed

Reviewer #2: All comments have been addressed

2. Is the manuscript technically sound, and do the data support the conclusions?

Reviewer #1: Yes

Reviewer #2: Yes

3. Has the statistical analysis been performed appropriately and rigorously? 

Reviewer #1: Yes

Reviewer #2: N/A

4. Have the authors made all data underlying the findings in their manuscript fully available?

Reviewer #1: Yes

Reviewer #2: Yes

5. Is the manuscript presented in an intelligible fashion and written in standard English?

Reviewer #1: Yes

Reviewer #2: Yes

6. Review Comments to the Author

Reviewer #1: Comment a:

The author really needs to clarify the modifications in the text in the form of color marking or highlighting, otherwise it is difficult to find the author's modifications, which will lead to rejection and unnecessary losses.

Comment b:

The author insists on taking "where to construct low-carbon housing" as the only competitive factor. The author needs to add at least one paragraph to explain that such assumptions have practical research significance and what potential problems exist in the assumptions, which will not have an important impact on the research value of the article.

Reviewer #2: Dear Author.

This manuscript has been properly revised in accordance with peer review comments.

Although the task may have been difficult due to the number of revisions, the content has been refined and appears to meet the standards of this journal.

I therefore recommend the Editor-in-Chief to publish this article in that journal.

Best regards.

7. PLOS authors have the option to publish the peer review history of their article (what does this mean?). If published, this will include your full peer review and any attached files.

Reviewer #1: **Yes: **Jiachen Fan

Reviewer #2: **Yes: **Yoshihiro Hamaguchi

---

## [Editor Report · Acceptance letter]

21 Mar 2023

PONE-D-22-27591R1 

Strategies of property developers in the context of carbon tax 

Dear Dr. Yao:

I'm pleased to inform you that your manuscript has been deemed suitable for publication in PLOS ONE. Congratulations! Your manuscript is now with our production department. 

Kind regards, 

on behalf of

Professor Bo Huang 

Academic Editor

PLOS ONE